# Examining the Role of Extraction Techniques and Regional Variability in the Antioxidant and Phytochemical Composition of *Juglans regia* L. Septa

**DOI:** 10.3390/plants14162524

**Published:** 2025-08-13

**Authors:** Jurgita Luksiene, Augusta Zevzikoviene, Jurga Andreja Kazlauskaite, Mindaugas Marksa, Daiva Majiene, Andrejus Zevzikovas

**Affiliations:** 1Department of Analytical and Toxicological Chemistry, Lithuanian University of Health Sciences, Sukileliu pr. 13, LT-50161 Kaunas, Lithuania; augusta.zevzikoviene@lsmu.lt (A.Z.); mindaugas.marksa@lsmu.lt (M.M.); andrejus.zevzikovas@lsmu.lt (A.Z.); 2Department of Drug Technology and Social Pharmacy, Lithuanian University of Health Sciences, Sukileliu pr. 13, LT-50161 Kaunas, Lithuania; jurga.andreja.kazlauskaite@lsmu.lt (J.A.K.); daiva.majiene@lsmu.lt (D.M.); 3Institute of Pharmaceutical Technologies, Lithuanian University of Health Sciences, LT-50161 Kaunas, Lithuania; 4Laboratory of Biochemistry, Neuroscience Institute, Lithuanian University of Health Sciences, Sukileliu pr. 13, LT-50162 Kaunas, Lithuania

**Keywords:** septa, walnuts, *Juglans regia* L., by-products, chemical composition, extraction, antioxidant potential

## Abstract

Walnut septa, traditionally discarded as waste in walnut processing because they primarily serve a structural function in the walnut fruit, have recently gained attention for their potential abundance of phenolic compounds, suggesting their overlooked value. This study aimed to optimise extraction parameters to maximise the extraction yield of bioactive compounds and explore regional variations in antioxidant activity and chemical composition of *Juglans regia* L. septa. The experimental variables included extraction methods (maceration, dynamic maceration, ultrasound processing, and reflux extraction), temperature, solvent type (methanol, acetone, and ethanol), and the percentage of water in the solvent. The optimal conditions were determined based on the total phenolic content—reflux extraction using 60% ethanol as a solvent for a duration of 60 min. Samples from 12 different regions in Lithuania, Armenia, and Ukraine were analysed for their phenolic and proanthocyanidin content and antioxidant activity using the CUPRAC method. The total phenolic content ranged from 131.55 to 530.92 mg of gallic acid equivalents per g of dry weight of plant material (mg GAE/g dw), while the proanthocyanidin content varied from 1.14 to 7.65 mg of (–)-epicatechin equivalents per g dry weight of plant material (mg EE/g dw). Among all the regions studied, the Šiauliai sample demonstrated the highest concentrations of phenolic compounds, proanthocyanidins, and antioxidant activity, with statistically significant differences compared to the other samples (*p* < 0.05). These findings demonstrate that walnut septa are a valuable source of phenolic compounds and antioxidants, with significant potential for developing natural nutraceuticals and antioxidant products.

## 1. Introduction

Walnuts (*Juglans regia* L.) have long been recognised for their nutritional and medicinal properties, primarily attributed to their high content of healthy fats, vitamins, and bioactive compounds [1]. However, a significant component of the walnut fruit, the septa or internal partitions, has been relatively underexplored in the scientific literature. Walnut septa, which serve as the structural support of the walnut, are typically considered a by-product and discarded during processing. Yet, emerging studies suggest that these internal partitions may be rich in phenolic compounds and could hold untapped potential as a source of bioactive molecules [2].

*J. regia*, commonly known as the English or Persian walnut, is a deciduous tree belonging to the family Juglandaceae. It is characterised by a broad crown, pinnate leaves, and large stone fruits. The plant thrives in temperate regions and is widely cultivated in areas such as Europe, North America, and parts of Asia. It prefers well-drained, fertile soils and is often grown in orchards for commercial nut production [3,4]. While the kernel is the most commonly consumed part, other parts such as the husk, shell, bark, and septa are also rich in bioactive compounds and have been explored for their medicinal and industrial uses. Several walnut-derived products are already available on the market, including walnut oil, extracts, and dietary supplements [3].

*J. regia* septum contains a distinct phytochemical profile that may offer notable therapeutic benefits [2,5]. Recent studies have demonstrated that it is abundant in various bioactive compounds, primarily polyphenols, including flavonoids and proanthocyanidins [6]. Flavonoids and proanthocyanidins are subclasses of polyphenols and are recognised for their role in combating oxidative stress; inhibiting cholesterol levels; and preventing lipid peroxidation, which has been implicated in the development of chronic diseases such as cancer, diabetes, and cardiovascular disease, as well as in improving vascular health and cognitive function [7,8]. Despite the known benefits of these compounds, the specific concentration and chemical profile of phenolic compounds in walnut septa remain understudied.

Walnut production is highly sensitive to cold temperatures during the winter and late spring, with the former posing a particular threat to *J. regia*, as it often results in early signs of tissue damage, such as bubbling [9]. Consequently, biological factors and climatic variability at varying altitudes play a crucial role in influencing both yield and quality. Cold stress can, however, induce protective physiological responses in walnut trees, leading to an increase in the synthesis of secondary metabolites, including phenolic compounds. These compounds, such as flavonoids and tannins, are believed to act as natural protectants by mitigating oxidative stress caused by low temperatures and enhancing cellular resilience. Such phenolic compounds also play a critical role in plant defence mechanisms, helping to protect against pathogens and environmental stressors [10,11]. Walnut trees are cultivated globally, particularly in temperate regions, where environmental conditions and climate fluctuations significantly affect their growth, productivity, and the accumulation of bioactive compounds [12].

The choice of extraction method and solvent highly influences the extraction of bioactive compounds from plant materials [13]. Common solvents used in polyphenolic extraction include ethanol, methanol, and acetone, or various solvent mixture systems, each with varying polarities and efficiencies for extracting different classes of compounds [14]. During the extraction of septa, ethanol is often preferred for its lower toxicity and effectiveness in extracting polar compounds [15,16]. Acetone and methanol were used in this study because of their proven ability to solubilise a wide spectrum of phenolic compounds, including both polar and semi-polar molecules. Methanol, in particular, is known for its efficiency in extracting high yields of phenolics, while acetone can complement this by dissolving compounds less soluble in alcohol-based solvents [17,18]. Although both methanol and acetone are acceptable within regulated limits, the use of these solvents requires careful monitoring. Acetone has been used less frequently in other similar studies, primarily in combination with other solvents to improve the solubility of bioactive compounds during extraction [5].

Similarly, the extraction method itself plays a critical role in determining the yield and quality of bioactive compounds. Traditional techniques, such as maceration and Soxhlet extraction, have been widely used [5,19]. Maceration involves soaking the plant materials in a solvent at room temperature for an extended period, while Soxhlet extraction uses continuous solvent reflux to ensure efficient extraction over several hours. However, both methods can be time-consuming and less solvent-efficient, meaning they require large amounts of solvent relative to the amount of extract obtained. Dynamic maceration involves the continuous stirring or agitation of plant materials with a solvent, enhancing solvent penetration and compound diffusion [20]. Reflux extraction, on the other hand, uses heat to cycle a solvent through the plant matrix repeatedly, allowing for thorough extraction over time [18]. More advanced techniques, such as ultrasound-assisted extraction (UAE), have gained popularity due to their enhanced efficiency and reduced extraction times [21,22]. UAE, which uses high-frequency sound waves to disrupt cell structures, has been shown to significantly increase the yield of bioactive compounds from walnut septum [23].

Much of the existing research on walnut phenolic compounds has focused on the kernels. There is a clear gap in the literature regarding the chemical composition and antioxidant activity of walnut septa. Moreover, walnut trees are cultivated in various geographical locations, like Lithuania, Ukraine, and Armenia, which could influence the chemical composition of their fruits due to climate and agricultural practices. Studies on geographical variation in the phytochemical content of nuts and fruits have revealed that environmental factors can significantly impact the nutritional and bioactive profiles of plant-based foods [24,25]. In terms of environmental and economic relevance, the septum accounts for approximately 2–3% of the total fruit weight, representing a considerable volume of agricultural waste [3]. This highlights the importance of studying walnuts from different regions to understand regional differences in chemical composition and antioxidant potential, which may contribute to the development of region-specific functional foods. To date, few studies have systematically explored the effects of extraction methods, solvent types, and regional differences in Lithuania, Ukraine, and Armenia on the bioactive profile of walnut septa. As a result, the full potential of walnut septa as a source of functional bioactive compounds remains underutilised.

This study aimed to optimise extraction parameters to maximise the yield of bioactive compounds and to investigate the influence of regional variations on the antioxidant activity and chemical composition of *J. regia* septa.

## 2. Results and Discussion

### 2.1. Determination of Phenols in Juglans regia L. Extracts

The extraction efficiency of phenolic compounds from walnut septa was evaluated under varying conditions: conventional maceration, dynamic maceration, UAE, and reflux. The solvents employed included acetone (A), ethanol (E), methanol (M), and water (W), each at varying concentrations (30–100%). Water was used as the diluent for all solvent preparations. Extraction durations of different times varied across the methods and were then tested.

#### 2.1.1. Extraction Using Maceration and Dynamic Maceration Methods

A comparative analysis reveals that dynamic maceration consistently yielded higher total phenolic concentrations across all solvents, concentrations, and time points compared to conventional maceration. Dynamic maceration is superior to simple maceration because it enhances solvent penetration and extraction efficiency through agitation on compound diffusion, consistent with previous findings in phenolic extraction studies on pomegranate peel [20]. It also offers better control over extraction parameters, making it more efficient for industrial-scale applications [20,26].

Among the solvents tested, acetone demonstrated the highest efficiency in extracting phenolic compounds using both methods of extraction. In particular, acetone at 50 to 70% concentrations achieved the highest yields (Figure 1 and Figure 2). The maximum phenolic content during conventional maceration was observed (270.70 ± 3.53 mg GAE/g dw) using acetone as a solvent at a 50% concentration after macerating the plant materials for 24 h (Figure 1).

Dynamic maceration yielded different results (Figure 2). During dynamic maceration, five samples showed the highest phenolic concentration and were statistically higher than all the other samples in all of the experiments. For this method, the best results were obtained using acetone (50%) and macerating for 6 h (290.20 ± 2.83 mg GAE/g dw), acetone (60%) and macerating plant materials for 6 and 12 h (285.65 ± 3.61 and 290.95 ± 3.18 mg GAE/g dw), and using ethanol (50%) for 24 h (293.14 ± 0.07 mg GAE/g dw).

During the extractions, the phenolic yield generally increased with a prolonged extraction time during both maceration methods. The most significant improvements were observed between 12 and 24 h, suggesting that longer extraction durations assist better compound diffusion and release from plant matrices. However, in a few instances, the yield decreased slightly at 24 h, which may indicate a few processes: saturation or degradation of the used solvent, physical changes in materials where prolonged exposure to the solvent may cause physical changes in active compounds, or oxidation due to light and oxygen exposure [27,28].

Although ethanol, particularly at a 70% concentration, also performed well, it was less effective than acetone. However, there was no statistical significance between these samples. Methanol showed variable results, with moderate extraction efficiency peaking at a 70–80% concentration. Water, in contrast, was consistently the least effective solvent, regardless of the extraction time or solvent, highlighting the importance of organic solvent polarity in facilitating the solubilisation of phenolic compounds [29].

In a study by Rusu et al., both maceration and ultra-turrax extraction were tested [5]. The results show that maceration using ethanol and acetone yielded a significantly lower total phenolic content compared to our study, using both methods. These differences may be attributed to differences in extraction parameters, such as extraction time and plant material, particle size, and even storage, as the concentration of phenols was relatively low.

#### 2.1.2. Extraction Using UAE Method

During the UAE technique, the extraction durations varied compared to maceration and were 10, 20, and 30 min, using the same solvents, acetone, ethanol, methanol, and water, at concentrations ranging from 30% to 100%.

Ultrasound was less effective compared to dynamic maceration but was more effective than conventional maceration to extract phenols from walnut septa. This may be due to the relatively short sonication time used or less efficient matrix disruption compared to thermal methods, as suggested by Shen et al. [30]. Acetone and ethanol were the most effective solvents in the UAE. Acetone showed promising results at a 40% and 70% concentration when the samples were processed for 30 min. The phenolic yields reached up to 253.85 ± 2.48 mg GAE/g dw (A 40%, 30 min) and 269.10 ± 5.80 mg GAE/g dw (A 70%, 30 min). Ethanol was the most effective at a 50% concentration, yielding 279. 35 ± 5.44 mg GAE/g dw after 30 min of ultrasound processing (Figure 3).

Methanol exhibited moderate performance, with optimal yields at concentrations of 50–70%. The highest yield observed was 230.85 ± 20.29 mg GAE/g dw (M 70%, 30 min), although slightly lower than that of acetone and ethanol under similar conditions. Water, in contrast, remained the least effective solvent, consistent with results from other extraction methods.

Increasing the extraction time generally led to higher phenolic yields across all solvent types. This is particularly notable in the 30 min extraction groups, where maximum values were consistently observed. The ultrasound mechanism likely enhances solvent penetration and matrix disruption, allowing for more efficient solute release with longer sonication [30].

#### 2.1.3. Extraction Using Reflux Method

Reflux extraction was evaluated as a thermal-assisted method for enhancing phenolic compound recovery from walnut septa. The extraction was carried out using the same solvents as mentioned before in previous methods, but the duration varied, including 30, 60, and 90 min.

The reflux yielded the best results in this study. Few studies have employed the reflux extraction method specifically for walnut (*J. regia*) septa. This improved efficiency is likely due to thermal softening of plant matrices and increased solubility of bound phenolic compounds, corroborating trends observed by Zhang et al. and Galanakis et al. [21,31]. Fourteen extracts showed the highest phenolic compound content, and although their values were statistically significantly higher compared to other groups, there were no significant differences observed among these top-performing samples themselves (Figure 4).

Among the solvents tested, acetone produced the highest phenolic yields, particularly at a 50, 60, and 70% concentration. When the heating was conducted for 90 min, the resulting concentration of phenolic compounds was 291.45 ± 2.47 mg GAE/g dw, 288.30 ± 6.93 mg GAE/g dw, and 290.70 ± 3.54 mg GAE/g dw. Ethanol also demonstrated high efficiency, particularly at 50–70% concentrations, with yields exceedingly approx. 290 mg GAE/g dw under extended durations. For example, ethanol at 70% for 90 min resulted in a phenolic content of approximately 290.70 ± 3.54 GAE/g dw, comparable to acetone’s performance. Methanol yielded moderately high phenolic concentrations, especially at 40–60%. Water, as observed in previous extraction methods, was the least effective solvent. Despite the extended extraction time and thermal assistance, phenolic concentrations remained below 218 mg GAE/g dw, emphasizing the inadequacy of water alone for polyphenol solubilization from walnut septa.

An increase in extraction time positively correlated with phenolic yield across nearly all the solvents. The most substantial increases were observed between 30 and 90 min, with 90 min treatments yielding peak phenolic concentrations in most cases. This time-dependent increase highlights the effect of prolonged thermal exposure and solvent saturation under reflux conditions, which facilitates the breakdown of plant matrices and enhances the release of bound phenolic compounds. However, for some combinations, particularly at very high solvent concentrations (like 100% acetone), yields declined or plateaued with time, likely due to compound degradation or the poor solubility of some phenolic classes in highly non-polar or polar solvents, as we saw in the maceration results as well [31].

One notable comparison can be made with a study by Elif Azize Özşahin Delibaş et al. [32], in which Soxhlet extraction using concentrated ethanol at 60 °C for 8 h yielded a total phenolic content of 119.42 ± 2.39 mg GAE/g dw. In our study, although the use of concentrated ethanol under reflux conditions resulted in lower phenolic yields, the values obtained were still considerably higher than those reported in many other investigations. This discrepancy may be attributed to several factors, including the geographical origin of the plant material, climatic variations, harvest time, and post-harvest storage conditions, all of which are known to influence the phytochemical composition of plant-derived extracts [33,34,35]. These findings are reported in other studies but not in walnut septa research, as it is limited.

Based on the results of all extraction methods, the most cost-effective conditions for extracting phenolic compounds from walnut septa the reflux extraction was chosen for further research. The selected parameters were as follows: 60 min using 60% ethanol.

#### 2.1.4. Regional Variations in *J. regia* Septa on the Phenols’ Content

After choosing the right extraction conditions, plant materials were collected from different regions of Lithuania and a few countries, like Armenia and Ukraine.

The Šiauliai sample demonstrated the highest value (530.92 ± 2.03 mg GAE/g dw), followed by those from Utena, Veliuona, and Gargždai, all exceeding 380 mg GAE/g. The lowest concentrations were recorded in samples from Anykščiai district and Armenia, which were 131.55 ± 0.59 and 193.85 ± 1.36 mg GAE/g dw, respectively. The highest results obtained were significantly greater, even when compared with the samples used to determine the optimal extraction conditions (Figure 5). These results suggest that Lithuanian-grown walnut septa, particularly from northern and central regions, may serve as superior sources of bioactive phenolic compounds.

Šiauliai, located in northern Lithuania, exhibits cooler average temperatures. Lower temperatures have been shown to trigger phenolic synthesis in plant tissues as a defence mechanism, as reported in other studies of various plants [4,36,37].

### 2.2. The Determination of the Total Proanthocyanidin Content

The proanthocyanidins showed similar results as phenolic compounds (Figure 6). The highest levels were detected in samples from Šiauliai and Gargždai 7.65 ± 0.05 and 7.09 ± 0.04 mg EE/g dw. Once again, the results obtained from walnut septa from Armenia and Ukraine were quite low. Nevertheless, the lowest result in all of the experiments was found in a sample collected from Anykščiai district. The concentration of proanthocyanidins was merely 1.14 ± 0.02 mg EE/g dw.

The strong correlation between phenolic compound content and proanthocyanidins can be explained by the chemical nature of proanthocyanidins. Proanthocyanidins are a subclass of flavonoids and therefore contribute directly to the total phenolic content [38].

While total phenolic measurements include a broad spectrum of compounds, the proportional relationship observed here implies that proanthocyanidins are a significant contributor to the overall polyphenol profile in walnut septa. As there is very limited research on the total proanthocyanidin content, we cannot compare our results with those reported in the literature.

### 2.3. Regional Variations in Antioxidant Activity

The CUPRAC method was chosen to assess the antioxidant activity in walnut septa due to its ability to measure both hydrophilic and lipophilic antioxidants under near-neutral pH conditions, therefore preserving sensitive compounds and offering greater sensitivity [39]. The study demonstrated strong positive correlations between antioxidant activity and both the total phenolic content (r = 0.97) and proanthocyanidin levels (r = 0.99) in walnut septa extracts. Additionally, the total phenolic content and proanthocyanidins were highly correlated with each other (r = 0.95), confirming that these compounds are major contributors to antioxidant capacity.

The obtained results show that the sample prepared from walnut septa collected in Šiauliai had the highest antioxidant capacity (1503.26 ± 4.61 μmol TE/g dw) (Figure 7). This result was statistically significantly higher compared to all other obtained antioxidant activity results. High values were also noted in the samples from Utena, Gargždai, and Šakiai district. The consistency between antioxidant activity and phenolic content supports the reliability of phenolics as key bioactive contributors to antioxidant capacity [40]. Conversely, walnut septa from Armenia and Ukraine exhibited notably lower antioxidant values, which may reflect differences in growing conditions, genotype, or post-harvest processing.

Overall, the results highlight the significant influence of growing region on the antioxidant profile of walnut septa. Extracts from cooler climates, particularly Šiauliai, may represent more potent sources of natural antioxidants for applications in functional foods, nutraceuticals, or cosmetics.

### 2.4. Chemical Composition of the Juglans regia L. Septa

An LC-MS-PDA analysis of *J. regia* septa extract samples was performed, and the determined compounds were compared to the published data of walnut septa profiles [7,17,41]. Nine phenolic compounds were identified (Figure 8), including ellagic acid, quercitrin, and caffeoyl derivatives, which are widely recognized for their antioxidant and anti-inflammatory properties. The presence of these compounds supports the high antioxidant capacity observed in the CUPRAC assay and reinforces the potential of walnut septa as functional ingredients in nutraceuticals and natural antioxidant products [6,7].

Although samples from several regions were collected, an LC-MS analysis was performed on just one representative extract, since the compound profiles were very similar and had no statistically significant difference.

It was determined that peak 1 (*m*/*z* 783 [M-H]−) was pedunculagin/casuarrin isomer (bis-HHDPglucose) 1, and peak 2 (*m*/*z* 633 [M-H]−) was a strictinin/isostrictinin isomer (galloyl-HHDP-glucose). Peak 3 (*m*/*z* 783 [M-H]−) was pedunculagin/casuarrin isomer (bis-HHDPglucose) 2. Furthermore, two isomers of tellimagrandin were identified: peak 4 (*m*/*z* 785 [M-H]−) was isomer 1, while peak 5 (*m*/*z* 785 [M-H]−)—isomer 2. Peak 6 (*m*/*z* 433 [M-H]−) was ellagic acid pentoside, peak 7 (*m*/*z* 301 [M-H]−) was ellagic acid, peak 8 (*m*/*z* 447 [M-H]−) was quercitrin, and peak 9 (*m*/*z* 487 [M-H]−) was caffeoyl hexoce-deoxyhexoside. The results of the LC-MS-PDA analysis are presented in Table 1.

## 3. Materials and Methods

### 3.1. Plant Materials and Reagents

Walnut (*J. regia*) septa were collected from various regions of Lithuania (Šiauliai (56.07331, 23.46924), Šakiai district (54.86087, 23.07875), Širvintos (55.05227, 24.94420), Utena (55.50453, 25.59484), Marijampolė (54.54829,23.37683), Lazdijai (54.22736, 23.51041), Kaunas (54.84244, 24.02246), Veliuona (55.07759,23.27818), Gargždai (55.70520, 21.39497), Biržai (56.19480, 24.75569), Anykščiai district (55.50758, 24.74563), and Alytus (54.38703, 23.96509), as well as Armenia and Ukraine. The walnut septa used for optimizing extraction conditions were from the last year’s harvest in Kaunas, which was a blend from various Kaunas places. Ethanol (96%), used for extraction, was purchased from Vilniaus degtine (Vilnius, Lithuania). In this experiment, purified water and deionized water were prepared with GFL2004 (GFL, Burgwedel, Germany) and millipore, SimPak 1 (Merck, Darmstadt, Germany).

DMCA reagent (4-dimethylaminocinnamaldehyde), (–)-epicatechin, formic acid, gallic acid, acetic acid, copper (II) chloride, an ammonium–acetate buffer, Trolox (6-Hydroxy-2,5,7,8-tetramethylchroman-2-carboxylic acid), and neocuproin were obtained from Sigma-Aldrich (St. Louis, MO, USA). Acetonitrile, methanol, and acetone was obtained from Fisher Scientific (Waltham, MA, USA). Folin–Ciocalteu’s phenol reagent was obtained from Merck (Darmstadt, Germany).

### 3.2. Extraction Conditions

#### 3.2.1. Maceration

Walnut septa were extracted using a maceration technique at room temperature (25 ± 2 °C). A sample of dried and milled plant materials (the material was milled using a 0.25 mm mill sieve) of 0.1 ± 0.001 g of septa was macerated in 10 mL of solvent. Different solvents (water (W), acetone (A), ethanol (E), and methanol (M)) were employed. The extractions were performed using different time intervals: 6 h, 12 h, and 24 h. After the extraction, the samples were centrifuged (Centurion Scientific, model C206, Marlborough, UK) (for 10 min at 3382× *g*), filtered with a paper filter (0.22 µm pore size), and stored in a refrigerator at 4 °C until further analysis. No solvent evaporation step was applied post-extraction, since analyses were conducted immediately after filtration and centrifugation.

#### 3.2.2. Dynamic Maceration

For dynamic maceration, the Elpan laboratory shaker type 358S, Poland, was used. The flasks were wrapped in aluminium foil and sealed with stoppers. The extraction parameters were the same as those used in Section 3.2.1.

#### 3.2.3. Ultrasound-Assisted Extraction (UAE)

Ultrasound-assisted extraction was performed using an ultrasound bath (frequency of 38 kHz) (Grant Instruments™ XUB12 Digital, Cambridge, UK). The amount of plant materials and solvents used were the same as per previous methods (Section 3.2.1.). The processing time varied from 10 min to 30 min. After the extraction, the samples were cooled down at room temperature (25 ± 2 °C), then centrifuged (Centurion Scientific, model C206, UK) (for 10 min at 3382× *g*), filtered with a paper filter (0.22 µm pore size), and stored in a refrigerator at 4 °C until further analysis. No solvent evaporation step was applied post-extraction, since analyses were conducted immediately after filtration and centrifugation.

#### 3.2.4. Reflux Extraction

Amounts of 0.1 ± 0.001 g of dried and milled plant materials were mixed with 10 mL of a solvent (the same solvent used in previous methods) in a round-bottom flask and refluxed in a glycerol bath (Heidolph Instruments GmbH&Co. KG, type heizbad Hei-VAP, Schwabach, Germany) for 30 min to 90 min. After that, the mixture was left to cool at a temperature of 25 ± 2 °C. The samples were centrifuged (Centurion Scientific, model C206, UK) (for 10 min at 3382× *g*), filtered with a paper filter (0.22 µm pore size), and stored in a refrigerator at 4 °C until further analysis. No solvent evaporation step was applied post-extraction, since analyses were conducted immediately after filtration and centrifugation.

### 3.3. Determination of Phenolic Compounds Using Spectrophotometry

A 1 mL aliquot of the extract was combined with 1 mL of Folin–Ciocalteu’s reagent and 9 mL of distilled water. After 5 min, 10 mL of a 7% sodium carbonate solution was added, and the volume was adjusted to 25 mL with distilled water. The mixture was then incubated in the dark at room temperature for 90 min. Following the incubation period, the absorbance of the solution was measured at 750 nm. The total phenolic content was determined using a gallic acid (0.0613–2 mg/mL) calibration curve (y = 0.9068x + 0.0617; R^2^ = 0.9996) and is expressed as mg of gallic acid equivalents per g of dry weight of plant material (mg GAE/g dw) [42].

### 3.4. DMCA Colorimetric Method for Quantification of Proanthocyanidins

A 200 µL aliquot of the extract was combined with 3 mL of a 0.1% DMCA reagent solution in acidified ethanol (9:1, *v*/*v*). The reference solution consisted of acidified ethanol. After a 5 min incubation period, the absorbance of the mixture was measured at 640 nm. The total proanthocyanidin content was quantified using a calibration curve (0.0019–0.03125 mg/mL) for (–)-epicatechin (y = 32.301x + 0.0289; R^2^ = 0.9998) and is expressed as mg of (–)-epicatechin equivalents per g dry weight of plant material (mg EE/g dw) [43].

### 3.5. Measurement of Antioxidant Activity by CUPRAC Assay

The CUPRAC reagent solution was prepared by combining copper chloride (0.17 g/100 mL), neocuproin (0.1566 g/100 mL), and an ammonium–acetate buffer (pH 7) in equal volumes (1:1:1). The mixture was kept in the dark at room temperature for 30 min.

A total of 10 µL of sample extract was added to 3 mL of the CUPRAC solution, followed by a 30 min incubation period at room temperature. Absorbance was measured at 450 nm, and antioxidant activity is expressed as μmol of Trolox equivalents per g of dry weight of plant material (μmol TE/g dw) using the following calibration curve (1–32 μmol/mL): y = 0.00004x + 0.0215 (R^2^ = 0.9991) [44].

### 3.6. Chemical Composition Determination Using LC-MS

A qualitative analytical profile of the *J. regia* septa extract sample was determined by performing an HPLC analysis in combination with mass spectrometry (MS) on the extract sample. The analysis was performed using the extracts obtained from the optimized reflux extraction method, which involved a 60 min extraction period with 60% ethanol. For compound separation, reverse-phase liquid chromatography (RP-LC) was used with a Photodiode Array (PDA) detector for UV-Vis detection. A mass spectrometry analysis was performed using electrospray ionization (ESI) in both negative and positive modes. The LC-MS system composed of a Shimadzu Nexera X2 LC-30AD HPLC system (Shimadzu, Tokyo, Japan) equipped with an LCMS-2020 mass spectrometer (Shimadzu, Tokyo, Japan).

The chromatographic separation was performed using a YMC-Triart C18 (YMC Karasuma-Gojo, Kyoto, Japan) (150 mm × 3.0 mm, 3 μm) analytical column, and its temperature was set to 25 °C.

Water and acetonitrile were used as mobile phase A and mobile phase B, respectively, both consisting of formic acid (0.1% concentration in both solutions), acting as an ion-pairing agent. For each sample, 10 μL aliquot volumes were injected into the chromatographic column. The gradient elution profile was as follows: 0.01–8.0 min, 5–15% B; 8–30 min, 15–20% B, 30–48 min, 20–40% B; 48–58 min, 40–50% B; 58–65 min, 50% B; 65.0–66.0 min, 50–95% B; 66–70 min, 95% B; 70–71 min, 95–5% B; 71–76 min, 5% B. The eluent flow rate in the HPLC analysis was 500 μL/min.

The optimum ESI conditions were set as 350 °C for the interface temperature, 250 °C for the DL temperature, 400 °C for the heat block temperature, 1.5 L/min for the nebulising gas flow, and 10 L/min for the drying gas flow. Positive-ion and negative-ion measurements were performed while switching alternately between the positive and negative ionisation modes. The *m*/*z* ranges for the positive and negative modes were 50–2000 *m*/*z* (mass-to-charge ratio). The scan speed in negative mode was 15.000 μ/s and 5.000 μ/s for positive mode. A step size of 0.1 *m*/*z* was used.

### 3.7. Statistical Analysis

Data were analysed using SPSS version 20.0 (IBM Corporation, Armonk, NY, USA). The total phenolic content, proanthocyanidins, and CUPRAC method experiments were performed three to five times ((*n* = 3) or (*n* = 5)). Data are expressed as the mean ± standard deviation (S.D.). Comparisons between three different measurements were made using the Friedman and Wilcoxon tests. In addition, comparisons between the two groups were made using the Mann–Whitney U test. A ranking of extraction groups based on the total phenolic content was conducted. The numbers at the top of each column (1–105) represent extraction groups, ranked by their cumulative performance across measured parameters. Higher numbers indicate better extraction efficiency and greater statistical significance compared to lower-ranked groups. The results were considered statistically significant at *p* < 0.05.

## 4. Conclusions

This study demonstrated that reflux extraction with 60% ethanol for 60 min was the most efficient method for isolating phenolic compounds from walnut septa, producing significantly higher yields compared to maceration, ultrasound-assisted extraction, and other solvents tested. Although dynamic maceration yielded promising results, it required significantly longer processing times, making it less practical for efficient extraction. Other solvents, such as acetone and methanol, also showed good extraction efficiency under reflux conditions. However, their use is limited by the complexity of solvent removal and regulatory constraints, especially in pharmaceutical contexts where solvent residues must be strictly controlled.

Geographical origin was found to significantly influence the phytochemical composition and antioxidant capacity of walnut septa. Samples from Šiauliai exhibited the highest concentrations of phenolic compounds, proanthocyanidins, and antioxidant activity, with statistically significant differences (*p* < 0.05) compared to other regions. These results indicate a strong relationship between environmental conditions—such as cooler temperatures—and secondary metabolite accumulation.

The chemical profiling via LC-MS confirmed the presence of several key phenolics, including ellagic acid, quercitrin, and caffeoyl derivatives, indicating that walnut septa are a rich and diverse source of bioactive compounds.

This study confirms the value of walnut septa as a rich and underutilised source of bioactive compounds, supporting its potential in the development of nutraceuticals, functional foods, and natural antioxidants. As walnut septa account for approximately 2–3% of the fruit mass and are typically discarded, their valorisation presents both environmental and economic benefits.

## Figures and Tables

**Figure 1 plants-14-02524-f001:**
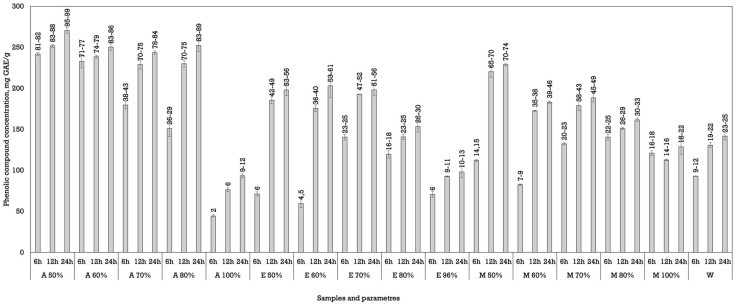
The phenolic compounds’ concentration in *Juglans regia* L. septa extracts using maceration. The data are presented as the mean ± S.D. (*n* = 3).

**Figure 2 plants-14-02524-f002:**
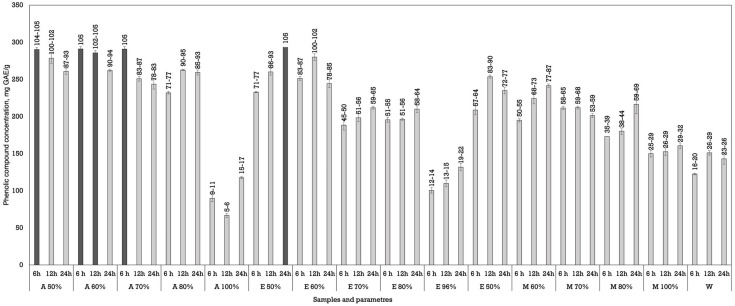
Extraction of phenolic compounds from *Juglans regia* L. using dynamic maceration. The data are presented as the mean ± S.D. (*n* = 3).

**Figure 3 plants-14-02524-f003:**
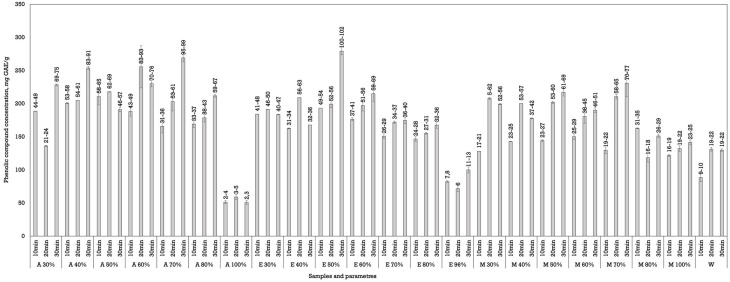
The effect of UAE duration and solvent concentration on phenolic content in septa extracts. The data are presented as the mean ± S.D. (*n* = 3).

**Figure 4 plants-14-02524-f004:**
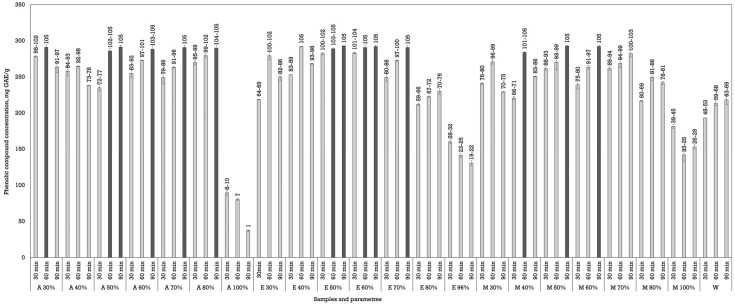
Impact of reflux extraction parameters on phenolic content in *Juglans regia* L. septa extracts. The data are presented as the mean ± S.D. (*n* = 3).

**Figure 5 plants-14-02524-f005:**
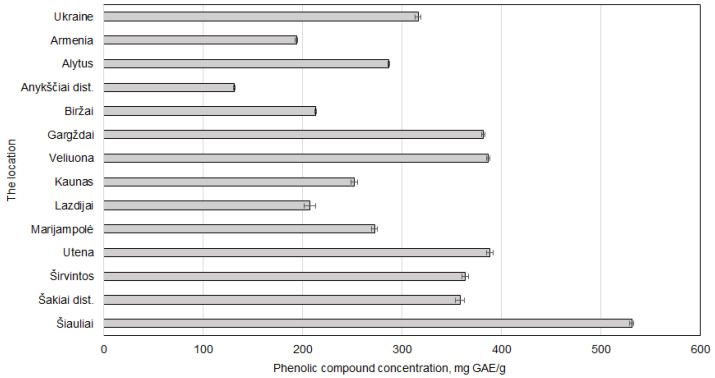
The regional variations in *Juglans regia* L. septa samples on the phenolic compound content. The data are presented as the mean ± S.D. (*n* = 5).

**Figure 6 plants-14-02524-f006:**
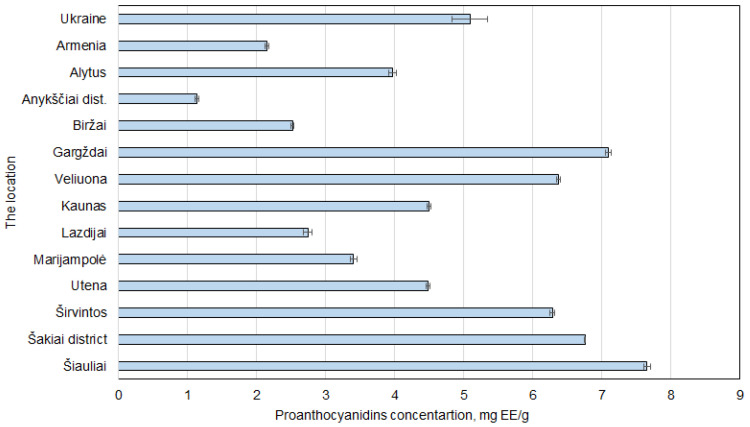
Regional differences in proanthocyanidin concentrations in *Juglans regia* L. septa samples. The data are presented as the mean ± S.D. (*n* = 5).

**Figure 7 plants-14-02524-f007:**
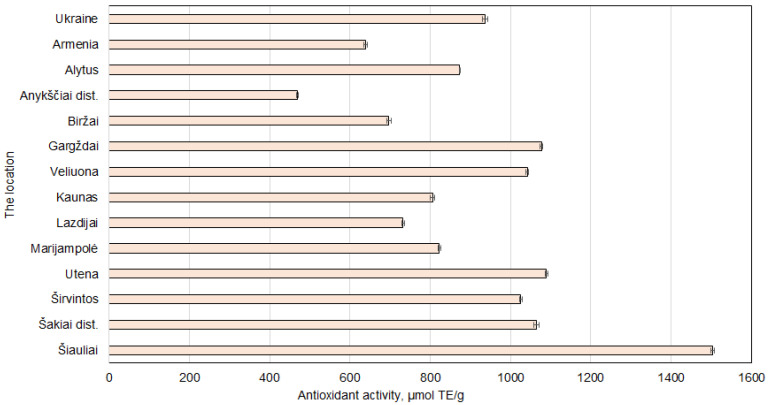
Antioxidant activity results for samples of *Juglans regia* L. septa collected from various locations. The data are presented as the mean ± S.D. (*n* = 5).

**Figure 8 plants-14-02524-f008:**
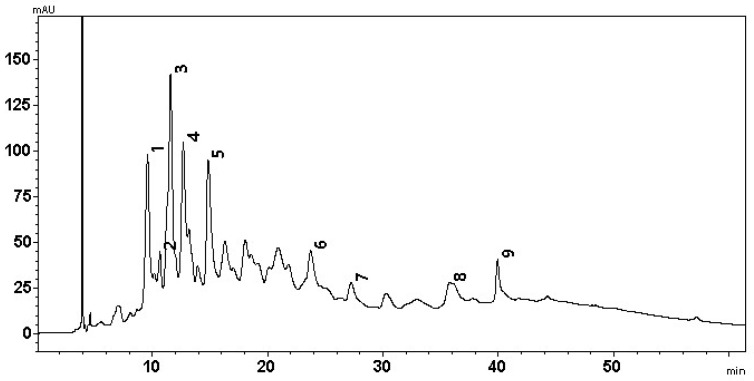
Chromatographic profile of walnut septa extract samples. The wavelength used to determine the compounds was 250 nm.

**Table 1 plants-14-02524-t001:** Identified compound list in walnut septa extract samples.

Compound	Retention Time (min)	[M-H]− (*m*/*z*)
Pedunculagin/casuarrin isomer (bis-HHDPglucose) (1)	9.598	783
Strictinin/isostrictinin isomer (galloyl-HHDP-glucose)	10.638	633
Pedunculagin/casuarrin isomer (bis-HHDPglucose) (2)	11.585	783
Tellimagrandin isomer (1)	12.691	785
Tellimagrandin isomer (2)	14.728	785
Ellagic acid pentoside	23.733	433
Ellagic acid	27.262	301
Quercitrin	35.657	447
Caffeoyl hexoce-deoxyhexoside	39.931	487

## Data Availability

Data are contained within the article.

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
