# Peer review of "Examining the Role of Extraction Techniques and Regional Variability in the Antioxidant and Phytochemical Composition of Juglans regia L. Septa"

_plants, 2025, doi:10.3390/plants14162524_

Round 1

Reviewer 1 Report

Comments and Suggestions for Authors

Abstract: line 21, it should be "extraction yield" not only "yield"; line 22, check unnecessary italics; line 27, it should be min instead of minutes; all abbreviations should be explained at the first mention in the abstract and again in the first mention in the Introduction. The use of aceton and methanol in nutraceuticals are absolutely unacceptable. Why did the authors employed them? Per g of dried extract or plant material? It should be defined in the abstract. The sentence 30-32 lines should be reformulated. The English language should be improved; the last sentence of the abstract is too general, thus, it should be changed, improved, and more specific for the presented research study.

Introduction: the full Latin name should be mentioned once and after that J. regia should be used; the plant should be explained in more detail (botanical characteristics, habitat, regions, plant parts which can be used, products on the market, etc.); lines 49-50, the mentioned compounds are also polyphenols; line 63, it should be "its"; the authors should point out hard reasons for the use of aceton and methanol as the extraction solvent, otherwise, it is for now the main disadvantages of the present study; line 66, it should be "obtaining" instead of "determing"; line 71, less solvent-efficient???; the authors did not explain dynamic maceration and reflux extraction; lines 76-88, the whole section should be transfered above (before the paragraph related to the extraction); the novelty of the present study should be more pointed out because it seems that there is a lack of novelty. Statistical data related to the amount of the waste as septa (in the included regions, as well as in the world) and costs/values should be provided in the Introduction; typing mistakes should be corrected. 

Results and Discussion: lines 109-110, non-complete sentence; lines 110-111, it is not clear-30% of water and 70% of...???; the whole 3.1. section is completely confusing; instead of hours should be h; all abbreviations in figures should be explained in their titles; the whole section of R&D should be significantly improved, more clear and better organized; in addition, deeper explanations of all obtained phenomenon, comparison to the literature data and similar studies, and new (recent) references should be provided in this section. Why did the authors measure only TPC for all extracts but not two other variables? Why was only one extract subjected to the HPLC analysis? The structural formulas of all identified compounds should be provided. The HPLC quantification of all main compounds should be provided in the manuscript with the aim to meet the quality criteria of the journal Plants.

Materials and Methods: line 309, it should be J.regia; the full name of Trolox (uppercase letter) should be provided; particle size of milled waste should be provided; "h" is missing after 12; the maceration process should be better explained (the name of the used shaker, speed, protection from the light, solvent evaporation, etc.); why did the authors use this solid to solvent ratio? The name of the centrifuge and its producer should be provided; line 329, the names of device and producer are missing, as well as the speed of rotation. Line 338, why was centrifugation time different? Since ultrasound waves cause destruction of plant material and produce smaller particle size of the plant material in comparison to the initial matrix, time of centrifugation should be prolonged. How did the authors control the temperature of the sample since US waves increase its temperature? In the abstract, the authors mentioned that they investigated the impact of water percentages in the extraction medium but from the sections 2.2., it is not clear. Reflux extraction should be explained in more detail. Section 2.3. - instead of minutes should be min; why did the authors prepare 25 mL of the mixture when 2 mL was enough for the spectrophotometric measurements? Did the authors prepare extracts in triplicates, as well as samples for the measurements? The authors should be consistent in writing unit for TPC, mg/mL or /g. How did the authors prepare dried extract if they expressed TPC per g of dried extract. The concentration range of gallic acid should be provided; the names of spectrophotometric device and producer should be provided; the same queries are for the section 2.4; the sentence should not start using numbers; why the authors did not use HPLC analysis for the quantification of the main compounds? The authors should explain in more detail how they identified compounds in the extracts.

Conclusion: the whole Conclusion section should be significantly improved and more specific and informative regarding the obtained results. Specific (not general as stated now) future perspectives, as well as further experiments should be mentioned. 

Latin names in the reference list should be in italics.

As far as I am concerned, the manuscript cannot be acceptable for publication without significant improvement (as stated above) and answering all above-mentioned queries. Therefore, my decision is Major Revision.

Reviewer 2 Report

Comments and Suggestions for Authors

Please differentiate somehow Augusta Zevzikoviene and Andrejus Zevzikovas in the Author Contributions part.

Line 18-19: „traditionally considered waste in walnut processing due to their structural role” – What do you mean by „due to their structural role”?

Line 64-65: „. Acetone in other studies was used rarely, unless in other solvent mixtures” – What do you mean by „unless in other solvent mixtures”?

Line 106, 107, 113, 159, 183, 230, 248, 267, 289, 308, 320, 321, etc. – Please, doublecheck the numbering of the subchapters.

Line 111: „acetone (A), ethanol (E), methanol (M), and water (W), each at varying concentrations (30–100%).” – Please define whether the dilutions were my by water? (Just to be sure.)

Figure 1: What do the numbers at the top of the columns mean? It would be worth writing the specific measurement values ​​here.

Figure 1, 3: It would be worth thickening the columns of the highest values ​​here as well, like in the other two figures.

Figures: Figures 3 and 4 also show 30 and 40% versions of each solvent, why is it different in Figures 1 and 2?

Line 188: „The 14 samples had the highest 188 phenolic compound content.” – Please specify the sentence.

Line 210: „This time-dependent shows” – a noun is missing

Line 106-266: What plant material was used for these measurements?

Line 308-311: It would be useful to provide a table with the most important parameters of the habitats (e.g. geographical coordinates, climate, precipitation, number of hours of sunshine) that could have influenced the amount of polyphenols.

Was the plant material collected from one or more trees in each habitat?

It would be good to clarify where in Ukraine and Armenia the sample came from (e.g. city, coordinates).

Line 297-304: It would be worth adding the number of components shown in Figure 8 in the table as well.

Line 312: „Ethanol (96 311 %) used for extraction was purchased from Vilniaus degtine (Vilnius, Lithuania). ” – Was it of analytical grade?

Line 322, 328: At what degree did the extraction occur?

Line 321: During maceration for 6-24 hs, were the bottles closed? Couldn't any solvent evaporate?

Line 337: „material and solvents were used the same as per previous methods.” – It would worth to mention the number of subchapter(s) like in Line 330.

Line 341: Which temperature did you use (because boiling points of the mentioned solvents are different)? Did you use a special equipment for this, like Soxhlet extractor?

Line 344: „After that, the mixture was left to cool at a temperature of 25 ± 2 °C” – A 30 minute ultrasonic water bath extraction also heats the extracts. Were those also cooled before filtration?

After maceration, the samples were centrifuged for 10 min, after the other techniques only for 5 min. Why it is different?

Line 345: Filtration with a paper filter (0.22 µm pore size) is missing from this subchapter.

Line 353, 359: What were the concentrations of gallic acid and epicatechin used for the preparation of calibration curves?

Line 371: The plant material, solvent, extraction time, method, filtration, etc. parameters are missing.

How many samples were analyzed using HPLC? Were the identified components found in each sample collected from different habitat? Or it was a pooled sample?

References: Please doublecheck this part, because some references are incomplete (e.g. 2, 9, 24)

Round 2

Reviewer 1 Report

Comments and Suggestions for Authors

Abstract:
line 25, chneck typing mistake (unneccesary itlaics in the word ''of'');
lines 29-30, the authors should mentioned why did they used those conditions (if they are optimal, it should be indicated);
lines 32-33, it should be mg gallic acid equivalents per g dry weight  (mg GAE/g dw) but the authors should specified the dry weight (plant material or dried extract); both comments are the same for mg EE/g;
line 44, ''traditionally considered waste'' should be deleted because it is aleredy written in the firts sentence of the abstract.

Introduction:  check typing mistakes in the whole manuscript (for example unnecessary italics of brackets,
line 57 - the Latin name of the plant should be in italics, the family name should not be in italics);
line 66, ''L.'' should be deleted;
lines 68-69, all mentioned compounds are contained in the previously mentioned one, i.e., polyphenols contain flavonoids and proanthocyanidins, while flavonoids contain proanthocyanidins. Therefore, the next sentence does not have a sence,because polyphenols are not subclasses of polyphenols (line 69); the sentence reletaed to the dynamic maceration should be before the sentence related to the UAE (line 99) which should be connected to the sentence in lines 105-107;
line 109, it shuld be J. regia (as abbreviation); the authors should explain what ''less solvent-efficient'' means;
lines 108-120, the whole section should be transfered above (before the paragraph related to the extraction); the novelty of the present study should be more pointed out because it seems that there is a lack of novelty. Statistical data related to the costs/values related to walnut septa should be provided in the Introduction; on line 139, the authors should use the abbreviation (as already stated in the first review process).
The authors did not answer all my queries in the section Introduction; thus, they cannot say that they have thoroughly improved the Introduction, according to the comments.

Results and Discussion: non-complete sentence, lines 143-144; The authors mentioned in the abstract that they investigate the impact of the temperature on the extract characteristics but there is no data related to it; the whole section of R&D should be significantly improved, clearer, and better organized; in addition, deeper explanations of all obtained phenomena, comparison to the literature data and similar studies, and new (recent) references should be provided in this section. The structural formulas of all identified compounds should be provided. The HPLC quantification of all main compounds should be provided in the manuscript to meet the quality criteria of the journal Plants. M&M; it cannot be ''mg/gmL per dry weight gallic acid equivalents'' mg GAE per g of dried plant material or dried extract; additionally, the authors did not answer all questions related to this section.

Conclusion: the authors did not improve this section; the results as values should not be in the conclusion section.

References: The authors wrote that the references was generated using zotero program, therefore they improved the sources where they could. This cannot be the excuse for an unacceptable reference list because they can prepare it mannually. As far as I am concerned, in the present form and without acceptance of all provided suggestions and implementation of the answered queries, requested analysis data (HPLC quantification data) and improvements, the manuscript does not meet the quality criteria of the journal Plants and cannot be considered for publication. Thus, my decision is Major Revision (again, since the authors did not answer many questions).

Round 3

Reviewer 1 Report

Comments and Suggestions for Authors

Abstract: Instead of ''minutes'' should be ''min''; the authors cannot write that the problem in presenting units is the problem with track changes - the query is clear - it should be ''mg of gallic acid equivalents per g of dry weight of plant material (mg GAE/g dw)'' and ''mg of (–)-epicatechin equivalents per g dry weight of plant material (mg EE/g dw)'' (the same in Materials and Methods section).

Introduction: line 145, the bracket should not be in italics; line 164, instead of ''primarily polyphenols, flavonoids, and proanthocyanidins'' should be ''primarily polyphenols, including flavonoids and proanthocyanidins'' because flavonoids and proanthocyanidins are polyphenols. Line 164, the authors wrote ''several studies'' but provided only one reference.

Results and Discussion: lines 562-563, the references are missing. Lines 674-675, the authors should provide statistical data as proof that a strong positive correlation exists between antioxidant activity, total phenolic content, and proanthocyanidin levels. The full Latin name of the plant should be used in the table titles.

References: In the Latin name of the plant, the first name is with an uppercase letter and the second is with a lowercase letter.

Conclusion: Instead of ''minutes'' should be ''min''; ''via'' should be in italics; the authors did not change all references according to the rules (the absence of italics in the Latin name of the plants, the absence of uniformity, for example, the absence of the abbreviation of the journal name, and uppercase letters in line 1169 and in the rest of the reference list). 

Author Response

Comment: Abstract: Instead of ''minutes'' should be ''min''
Response: We revised the abbreviations in the whole manuscript.

Comment: the authors cannot write that the problem in presenting units is the problem with track changes - the query is clear - it should be ''mg of gallic acid equivalents per g of dry weight of plant material (mg GAE/g dw)'' and ''mg of (–)-epicatechin equivalents per g dry weight of plant material (mg EE/g dw)'' (the same in Materials and Methods section)
Response: We fixed the mistake.

Comment: Introduction: line 145, the bracket should not be in italics
Response: We fixed the mistake.

Comment: line 164, instead of ''primarily polyphenols, flavonoids§, and proanthocyanidins'' should be ''primarily polyphenols, including flavonoids and proanthocyanidins'' because flavonoids and proanthocyanidins are polyphenols
Response: We fixed the mistake.

Comment: Line 164, the authors wrote ''several studies'' but provided only one reference
Response: We fixed the mistake.

Comment: Results and Discussion: lines 562-563, the references are missing
Response: While rearranging the paper, this sentence was unintentionally left out. As we adjusted the paragraphs and sentences, it has now been moved to its correct position.

Comment: Lines 674-675, the authors should provide statistical data as proof that a strong positive correlation exists between antioxidant activity, total phenolic content, and proanthocyanidin levels
Response: We included the inforation - the study demonstrated strong positive correlations between antioxidant activity and both total phenolic content (r = 0.97) and proanthocyanidin levels (r = 0.99) in walnut septa extracts. Additionally, total phenolic content and proanthocyanidins were highly correlated with each other (r = 0.95), confirming that these compounds are major contributors to antioxidant capacity.

Comment: The full Latin name of the plant should be used in the table titles
Response: We fixed the mistake.

Comment: References: In the Latin name of the plant, the first name is with an uppercase letter and the second is with a lowercase letter
Response: Thank you for your comment regarding the formatting of Latin plant names. We agree that, per botanical conventions, the genus should be capitalized and the species lowercase (e.g., Juglans regia). However, in the reference list, we have preserved the original formatting from the titles of the cited articles, where both parts are capitalized according to the journals’ style. As it is standard practice not to alter published titles in references, we have kept them as originally formatted.

Comment: Conclusion: Instead of ''minutes'' should be ''min''; ''via'' should be in italics
Response: We fixed the mistakes.

Comment: The authors did not change all references according to the rules (the absence of italics in the Latin name of the plants, the absence of uniformity, for example, the absence of the abbreviation of the journal name, and uppercase letters in line 1169 and in the rest of the reference list).
Response: We fixed the mistakes.
